# Combined Metabolome and Transcriptome Analysis Elucidates Sugar Accumulation in Wucai (*Brassica campestris* L.)

**DOI:** 10.3390/ijms24054816

**Published:** 2023-03-02

**Authors:** Chenggang Wang, Jiajie Zhou, Shengnan Zhang, Xun Gao, Yitao Yang, Jinfeng Hou, Guohu Chen, Xiaoyan Tang, Jianqiang Wu, Lingyun Yuan

**Affiliations:** 1College of Horticulture, Vegetable Genetics and Breeding Laboratory, Anhui Agricultural University, 130 West Changjiang Road, Hefei 230036, China; 2Provincial Engineering Laboratory for Horticultural Crop Breeding of Anhui, 130 West of Changjiang Road, Hefei 230036, China

**Keywords:** wucai (*Brassica campestris* L.), D-galactose, β-D-glucose, sugar accumulation pathway, interact network

## Abstract

Wucai (*Brassica campestris* L.) is a leafy vegetable that originated in China, its soluble sugars accumulate significantly to improve taste quality during maturation, and it is widely accepted by consumers. In this study, we investigated the soluble sugar content at different developmental stages. Two periods including 34 days after planting (DAP) and 46 DAP, which represent the period prior to and after sugar accumulation, respectively, were selected for metabolomic and transcriptomic profiling. Differentially accumulated metabolites (DAMs) were mainly enriched in the pentose phosphate pathway, galactose metabolism, glycolysis/gluconeogenesis, starch and sucrose metabolism, and fructose and mannose metabolism. By orthogonal projection to latent structures-discriminant s-plot (OPLS-DA S-plot) and MetaboAnalyst analyses, D-galactose and β-D-glucose were identified as the major components of sugar accumulation in wucai. Combined with the transcriptome, the pathway of sugar accumulation and the interact network between 26 DEGs and the two sugars were mapped. *CWINV4*, *CEL1*, *BGLU16*, and *BraA03g023380.3C* had positive correlations with the accumulation of sugar accumulation in wucai. The lower expression of *BraA06g003260.3C*, *BraA08g002960.3C*, *BraA05g019040.3C*, and *BraA05g027230.3C* promoted sugar accumulation during the ripening of wucai. These findings provide insights into the mechanisms underlying sugar accumulation during commodity maturity, providing a basis for the breeding of sugar-rich wucai cultivars.

## 1. Introduction

Wucai (*Brassica campestris* L. ssp. *chinensis var. rosularis* Tsen), a subspecies of non-heading Chinese cabbage, is widely grown in the Yangtze-Huai River Basin [1]. Wucai is rich in vitamin C, vitamin B1, and carotene, resulting in it being referred to as a “vitamin vegetable” [2]. Wucai leaves become sweet after undergoing autumn and winter growth, satisfying consumer preference due to their nutritional value and taste [3].

The sweetness of vegetables and fruit depends not only on the total amount of sugar but also on the sugar composition [4]. Sweetness is mainly conferred by sucrose, glucose, and fructose, which contribute differently to the sweetness of vegetables and fruit [4]. In Chinese cabbage, the leafy head is the storage organ and the internal midrib (IM) is the main tissue of sugar accumulation, which possesses the highest content of soluble sugar at harvest [5]. Fructose is the major sugar that accumulates in the internal tissues of Chinese cabbage, followed by glucose [5]. Differences in the sweetness of *Cucurbita* moschata were attributed to the content and composition ratio of sucrose [4,6]. As fructose tastes sweeter than sucrose and glucose, sucrose metabolism and the ratio of fructose/glucose were promoted in tomato fruits in order to improve the flavor quality [7,8]. The accumulation pattern and concentration of sugar vary with species and are regulated by fruit development [9]. Glucose is the main soluble sugar in mature pitaya fruit, whereas in ripened apricot fruits, glucose and sucrose are the major sugars [10,11]. The contents of sucrose, glucose, and fructose are high in harvested watermelon and mango fruit [12,13]. In melon fruit and sugarcane, sucrose was found to increase steadily with fruit development [14,15].

Sugar accumulation comes mainly from the transport of photosynthetic products, with sucrose being the form of transport in most plants, and a number of key enzymes can be involved in regulating sugar metabolism and, thus, the composition and content of sugars [16]. Sucrose phosphate synthase (SPS), one of the key enzymes in plant sucrose synthesis, catalyzes the production of sucrose as an irreversible reaction and is the rate-limiting enzyme for the synthesis of sucrose [17]. SPS activity is positively correlated with sucrose accumulation [18]. Transcript levels of *SPS* increased with sucrose accumulation during ripening in watermelon and banana [19,20]. In addition, the expression pattern of *SPS* in pineapple and potato all showed that its expression was related to sucrose metabolism [21,22]. Sucrose synthase (SUS) catalyzes both the breakdown of sucrose to UDP glucose and the synthesis of sucrose [17]. SUS is also one of the key enzymes for the entry of sucrose into various metabolic pathways, regulating the ability of the crop to metabolize sucrose and the amount of sucrose input [23]. During the development of apple fruit, with the accumulation of sucrose, the expression of *MdSUSY2*, *MdSUSY3*, and *MdSUSY4* decreased obviously, indicating that SUS mainly played a major role in the decomposition of apple sucrose [24]. The expression of *CitSus5* was increasing while that of *CitSus6* was gradually decreasing during fruit development in citrus, suggesting that SUS was involved in reversible reactions in citrus, possibly both synthesizing and breaking down sucrose [25]. Invertase (INV), also called sucrase, can hydrolyze sucrose into glucose and fructose. According to the site where the enzyme is present on the cell, INV mainly consists of cell wall convertase (CWINV), vesicle convertase (VINV), and cytoplasmic convertase (CINV) [26]. Based on the optimum *PH* of the enzyme, CWINV and VINV can be classified as acid convertase (AI), while CINV is a neutral invertase (NI) [26]. Numerous studies showed that there is a significant negative correlation between the activity of INV and sucrose accumulation in fruits [12,22,26,27]. In tomato fruits, CWINV and VINV are encoded by *LIN* and *VI*, respectively. *LIN5*, *LIN7*, *LIN8*, *LIN9*, and *VI* were upregulated by silencing *SWEET7* (Sugars Will Eventually be Exported Transporters) and *SWEET14* to increase CWINV and VINV activity [28]. It can be seen that the upregulation of *CWINV* and *VINV* increases the activities of AI, thereby promoting the hydrolysis of sucrose. CWINV is typically considered as a sink-specific enzyme, and its activity is usually low in source leaves [29]. However, both *MdCWINVs* (*MdCWINV2* and *MdCWINV3*) identified in apple had lower expression levels in the fruit than in the leaves, and the transcript levels of *MdCWINV2* and *MdCWINV3* declined dramatically during maturation [24]. Hexokinase (HK) could catalyze the phosphorylation of hexose, which could catalyze the conversion of glucose into glucose -6- phosphate (glucose -6P), and then enter the glycolytic pathway [30]. Overexpression of the *HK* was able to cause a significant reduction in the sugar content of plants [31].

Besides INV, SPS, and SUS, another enzyme related to sugar accumulation and metabolism in watermelon fruits is α -galactosidase [32]. Stachyose and raffinose are the main transportation forms of photosynthetic products in *Cucurbitaceae* plants, which can be decomposed by α -galactosidase to produce sucrose and galactose [33,34]. Cellulase (CL) is an important enzyme complex, mainly consisting of endoglucanase (EG), exoglucanase (CBH), and β-glucosidase (BGL), which hydrolyze cellulose to form glucose [35,36,37]. In previous studies, cellulose was considered to be related to the softening of crops during development [38,39]. Nevertheless, in biomass utilization, CL is employed to hydrolyze cellulose in multiple steps to generate glucose [40]. In studies of sugar accumulation in Chinese cabbage that is more closely related to wucai, it was noted that *BraA01gHT4* and *BraA03gHT7* were positively correlated with the soluble sugar content (mainly fructose and glucose) of the inner lobe, while *BraA03gFRK1*, *BraA09gFRK3*, *BraA06gSPS2,* and *BraA03gHT3* were negatively correlated with sugar content [41]. Furthermore, the high expression of *SUS1* was considered to promote the accumulation of fructose and glucose in leaf balls of Chinese cabbage [42]. Sweetness is a typical indicator and characteristic of maturation in wucai. In recent years, metabolomics (including liquid chromatography-tandem mass spectrometry (LC-MS/MS) and gas chromatography-tandem mass spectrometry (GC-MS/MS)) and transcriptomics (RNA sequencing (RNA-Seq)) have been successfully applied to reveal the mechanism of sugar accumulation in ripening fruits, such as Chinese cabbage, ponkan, and kiwifruit [41,43,44]. However, there are no studies that have reported on sugar accumulation during wucai maturation.

A biomarker is a characteristic biochemical index, which can be objectively measured to provide information about the biological process of the organism [45]. Metabonomics pays attention to the changes in small-molecule metabolites in organisms, which provides the possibility for identifying objective biomarkers. Scholars established and analyzed the OPLS-DA model or OPLS-DA-Splot map, and then potential biomarkers could be found in the project based on variable importance in the projection (VIP) score > 1 [46,47]. ROC (receiver operating characteristic curve) and AUC (area under ROC curve) diagnostics were performed using the online software MetaboAnalyst to identify potential biomarkers [46,47]. Combined analyses of the transcriptome and metabolome by LC-MS/MS and GC-MS/MS were conducted herein to investigate the molecular mechanism of sugar transformation in wucai during the maturation process, and the DAMs and related genes were identified. This is the first report on sugar biomarkers and the mechanisms of sugar accumulation with the maturity process of wucai. The results provide a valuable basis and reference for commercial applications and breeding programs for wucai.

## 2. Results

### 2.1. Changes in Sugar Content in the Wucai Leaves

A growth chamber was used for simulating the growth environment of wucai. To investigate the changes in soluble sugar content in wucai leaves during the growth period, we determined the soluble sugar content at nine sampling periods (Appendix A). The results showed that the soluble sugar increased gradually with the growth of wucai from 34 DAP and peaked at 46 DAP (Figure 1A). The time points of 34 DAP and 46 DAP were selected for the determination of D-galactose, glucose, fructose, and sucrose. It was found that the contents of D-galactose, glucose, and fructose increased significantly during the wucai maturation process (Figure 1B,C). The contents of D-galactose, fructose, and sucrose at 46 DAP were 1.40-, 1.16-, and 1.39-fold higher than those at 34 DAP, respectively (Figure 1B,D,E). There were significant differences in glucose between the two periods, reaching 3.58-fold (Figure 1C). Interestingly, at 46 DAP, the ratio of glucose/soluble sugar increased to 5.75% from 2.49% at 34 DAP, compared to D-galactose (Figure 1F,G). Therefore, we considered that these sugars, especially glucose, play vital roles in sugar transformation in wucai. According to the sugar change trend, 34 DAP and 46 DAP were selected as the two periods for further study.

### 2.2. DAM Analysis in Wucai Leaves

To further understand the changes in metabolites in the wucai leaves during sugar transformation, the metabolites at 34 DAP and 46 DAP were detected by LC-MS/MS and GC-MS/MS. The PCA of the metabolomic profiles of the 12 samples showed that the first principal component explained 46% (LC-MS/MS) and 47.2% (GC-MS/MS) of the total variance and distinguished the samples based on the two periods (34 DAP and 46 DAP) (Appendix A). A total of 650 and 111 DAMs were identified with *p* < 0.05 and VIP >1 from the LC-MS/MS and GC-MS/MS analysis, respectively (Appendix A). In the LC-MS/MS analysis, compared to 34 DAP, a total of 385 DAMs were upregulated (fold change, log_2_(FC) > 0) and 265 DAMs were downregulated (log_2_(FC) < 0) at 46 DAP (Appendix A). The proportion of organooxygen compounds/total DAMs was 12.923%, which was the maximum in any class category (Appendix A). The organooxygen compounds mainly included 66 carbohydrates and carbohydrate conjugates, nine phenols and polyols, six carbonyl compounds, and three ethers (Figure 2A). The proportion of carbohydrates and carbohydrate conjugates/total DEMs accounted for 10.15%, which was significantly higher than those of the other metabolites according to the sub-class category (Appendix A). The metabolomic analysis showed that the DAMs were mainly enriched in carbohydrates and carbohydrate conjugates. There were 35 upregulated and 31 downregulated DAMs (Figure 2B). The GC-MS/MS analysis showed that there were 43 upregulated and 68 downregulated DAMs at 46 DAP compared to at 34 DAP (Appendix A). Only 13 DAMs (seven upregulated and six downregulated) were classified as carbohydrates and carbohydrate conjugates (Appendix A and Figure 2C).

### 2.3. Kyoto Encyclopedia of Genes and Genomes (KEGG) Enrichment Analysis of DAMs Related to Sugar Accumulation in Wucai Leaves

To identify the major pathways of DAMs related to sugar accumulation in wucai leaves, KEGG enrichment analysis was conducted. The *p*-value in the pathways indicates the significance and Rich factor derived from ratio of DAMs/total metabolite number in the pathway. The LC-MS/MS analysis showed that DAMs related to sugar accumulation were notably enriched in the pentose phosphate pathway (ath00030), galactose metabolism (ath00052), glycolysis/gluconeogenesis (ath00010), and fructose and mannose metabolism (ath00051) pathways (Figure 3A). In the GC-MS/MS analysis, DAMs related to sugar accumulation were notably enriched in galactose metabolism (ath00052) and starch and sucrose metabolism (ath00500) (Figure 3B). It was interesting that the enrichment pathways in LC-MS/MS and GC-MS/MS were somewhat distinct. The reason could be due to the variation in quantities of other DAMs detected in the LC-MS/MS and GC-MS/MS analyses.

### 2.4. Biomarkers Analysis Related to Sugar Accumulation

In the DAM analysis, we found that many carbohydrates and carbohydrate conjugates were upregulated. However, the major sugars involved in sugar accumulation in wucai were still unclear. OPLS-DA, a supervised discriminant analysis statistical method, was used to intuitively identify the differences between samples. The VIP score was obtained according to the OPLS-DA model, and potential biomarkers were distinguished with VIP > 1. We found that the numbers and fold-change of the DAMs related to sugar accumulation in the GC-MS/MS analysis were generally lower than those of the LC-MS/MS analysis. Consequently, OPLS-DA S-plot analysis based on the LC-MS/MS data was performed to identify significant DAMs and potential biomarkers. A total of 17 DAMs identified as biomarker candidates were filtered in the OPLS-DA S-plot (Appendix A and Appendix A). Of the candidates, the differential accumulation of β-D-glucose, D-galactose, and trehalose was significant (Appendix A). β-D-glucose and D-galactose, which are carbohydrates and carbohydrate conjugates, were upregulated (Appendix A). In order to more rigorously assess the results and their accuracy, further analysis of biomarkers was conducted using MetaboAnalyst 5.0 (https://www.metaboanalyst.ca/, accessed on 27 August 2021). Thirteen biomarkers were screened based on log_2_(FC), t-tests, and AUC (Table 1). This showed that β-D-glucose and D-galactose had excellent AUC and log_2_(FC) values (Figure 4). The result validated that β-D-glucose and D-galactose could indeed be the major sugars in sugar accumulation in wucai and had positive effects on the sweetness. D-galactose also participates in amino sugar and nucleotide sugar metabolism (ath00520), and this pathway was screened for further analysis. We found that DAMs involved in enrichment pathways in the GC-MS/MS analysis were also present in the LC-MS/MS data. Thus, the DAMs in the pentose phosphate pathway (ath00030), galactose metabolism (ath00052), glycolysis/gluconeogenesis (ath00010), fructose and mannose metabolism (ath00051), starch and sucrose metabolism (ath00500), and amino sugar and nucleotide sugar metabolism (ath00520) were analyzed by making a heatmap based on the LC-MS/MS data (Figure 5A). The metabolites that accumulated significantly in these enrichment pathways were D-glycoldehyde3-phosphate, D-fructose, D-(+)-raffinose, Galactonic acid, N-acetyl-D-glucosamine, β-D-fructose 6-phosphate, β-D-Glucose, Gluconolactone, Fucose 1-phosphate, levan, and Glucose 6-phosphate.

### 2.5. Transcriptome Analysis

Six mixed replicates of wucai leaves at two periods (34 DAP and 46 DAP) were subjected to RNA-Seq analysis in order to identify the potential molecular mechanisms responsible for sugar accumulation in wucai. After filtering, a total of 39.70 G of clean data were obtained from the wucai leaves. The Q30 (sequences with sequencing error rates lower than 0.1%) content of the six cDNA libraries were more than 92.63%, and the average GC content was 48.07% (Appendix A). Overall, the data indicated that the Illumina sequencing data were of high quality and could be used for further analysis (Appendix A).

All 4761 unigenes were searched in the Gene Ontology (GO) and KEGG databases, with 3431 and 1110 corresponding annotated unigenes. The GO term analysis of the wucai leaf transcriptome showed that 21 terms were related to the biological process category, of which “biological regulation,” “cellular process,” “metabolic process,” and “single-organism process” were the main GO terms (Appendix A). Thirteen terms were correlated with the cellular component category, of which “cell,” “cell part,” and “organelle” were the most abundant GO terms. Twelve terms were included in the molecular function category, of which “binging” and “catalytic activity” made major contributions. In addition, 18 KEGG pathways were annotated, among which “carbohydrate metabolism,” “translation,” and “signal transduction” were the most abundant KEGG pathways (Appendix A).

### 2.6. Coexpression Analysis of Genes Related to D-Galactose and β-D-Glucose Accumulation

The major sugars involved in sugar metabolism in wucai are D-galactose and β-D-glucose. To explore the metabolic differences in the two sugars at the sugar transformation periods, the accumulation of the two sugars was analyzed by combined transcriptome and metabolome analysis. D-galactose and β-D-glucose were mainly involved in the galactose metabolism (brp00052), glycolysis/Gluconeogenesis (brp00010), and starch and sucrose metabolism (brp00500) pathways, and, thus, we focused on DEGs related to these three metabolic pathways. It was found that most genes related to starch degradation and synthesis, trehalose synthesis, and phosphorylating D-fructose, D-glucose, and β-D-glucose were downregulated (Figure 5B). The downregulated DEGs mainly included *BAM* (β-amylase), *DPE* (4-alpha-glucanotransferase), *PHS* (α-glucan phosphorylase), *SS* (starch synthase), *SBE* (1,4-alpha-glucan-branching enzyme), *TPS* (α-, α-trehalose-phosphate synthase), *TPP* (trehalose-phosphate phosphatase), and other genes (hexokinase).

According to the major two sugars and related DEGs, we constructed an accumulation pathway of D-galactose and β-D-glucose (Figure 6A). In this way, there were three DEGs encoding INV, namely *CWINVs* (*CWINV3*, *CWINV4*) and *VINV*(*BRFUCT3*), all of which encode AI. Of these genes, only the expression of *CWINV4* was up-regulated. Raffinose and stachyose located in the galactose metabolic pathway were decomposed into D-galactose under AI (*CWINV4*). In the meantime, raffinose and stachyose were hydrolyzed into D-glucose under the action by the same gene. *CWINV4* was also present in the starch and sucrose metabolic pathway, converting sucrose to D-glucose by hydrolysis. Moreover, cellulose in the starch and sucrose metabolic pathway was hydrolyzed to generate D-glucose. There were six DEGs associated with cellulose hydrolysis, EG (*BraA03g023380.3C*, *CEL1*) was up-regulated, while only one (*BGLU16*) of the BGL DEGs (*BGLU16*, *BGLU9*, *BGLU15*, and *BGLU47*) was up-regulated. Under the synergistic effect of *BraA03g023380.3C*, *CEL1*, and *BGLU16*, cellulose was gradually hydrolyzed into D-glucose. Aldose 1-epimerase (AEP) was able to catalyze the conversion of D-glucose to β-D-glucose. The generated D-glucose was converted to β-D-glucose by up-regulated expression of *ARB_05372* (AEP). HK could phosphorylate β-D-glucose to β-D-Glucose 6-phosphate (β-D-glucose 6P), which later entered the glycolysis pathway. The four HK DEGs identified in this paper (*BraA06g003260.3C*, *BraA08g002960.3C*, *BraA05g019040.3C*, and *BraA05g027230.3C*) were all down-regulated, reducing the phosphorylation of β-D-glucose and promoting the accumulation of the sugar. The genes (galactokinase) catalyzing D-galactose were not differentially expressed, which showed that the accumulation of D-galactose mainly depended on AI under the action of *CWINV4* during the maturation process of wucai. In the transcriptome analysis, the FPKM value of *CWINV4* at 34 DAP was zero. Hence, the relative expression of *CWINV4* in the roots, stems, leaves, and petioles at 34 DAP and 46 DAP was detected. The relative expression of *CWINV4* at 46 DAP was generally higher than that at 34 DAP in the four tissues, especially in the leaves and petioles (Appendix A).

To explore other genes that contribute to the accumulation of D-galactose and β-D-glucose, we selected TOP100 DEGs in the transcriptome and calculated the correlation between the expression of DEGs and response intensity data of biomarkers using the Pearson correlation method. DEGs with correlation values ≥0.98 or ≤−0.98 and *p* < 0.05 were selected and an interaction network was produced (Figure 6B). These were 26 and 8 DEGs that were significantly associated with D-galactose and β-D-glucose, respectively. *BraA09g036850.3C* and *BraA01g000700.3C* had a significant positive correlation with both D-galactose and β-D-glucose (Figure 6B). The DEGs with a significant negative correlation with β-D-glucose were *SAHH2 (adenosylhomocysteinase 2), CHI (chalcone-flavonone isomerase), CHS1 (chalcone synthase 1), CHS3 (chalcone synthase 3-like), FLS1 (flavonol synthase/flavanone 3-hydroxylase), and OMT1 (flavone 3’-O-methyltransferase 1-like)*, which also had a significant negative correlation with D-galactose (Figure 6B).

### 2.7. Changes in Relative Expression Levels of DEGs and Enzyme Activities

Twenty DEGs in the KEGG pathways and eight DEGs significantly associated with both D-galactose and β-D-glucose were selected and we measured their relative expression levels at 34 DAP, 37 DAP, 40 DAP, 43 DAP, and 46 DAP (Figure 7). The changes in the relative expression level of these genes at 46 DAP vs. 34 DAP were consistent with the transcriptome data (Figure 6 and Figure 7). The relative expression levels of *CWINV4*, *BraA03g023380.3C*, *BGLU16,* and *ARB_05372* showed an increasing trend from 40 DAP and peaked at 46 DAP (Figure 7). *CWINV3*, *BGLU9*, *BGLU15*, *BGLU47*, *BraA06g003260.3C*, *BraA05g027230.3C*, *BraA05g019040.3C*, *BAM1*, *BAM3-like*, *SAHH2*, *CHI,* and *FLS1* were genes that were down-regulated in the transcriptome, generally peaking at 37 DAP or 40 DAP and continuing to be downregulated until 46 DAP (Figure 7). *CEL1*, *SUS3,* and *BraA01g000700.3C* had irregularly varying relative expression levels, but the highest expression was observed at 46 DAP (Figure 7). Although *BraA09g036850.3C* was upregulated around maturation, its expression level peaked at 37 DAP (Figure 7). These results suggested that these DEGs may function at different stages.

To understand whether the enzymes encoded by these genes play a role in sugar accumulation, we determined eight enzyme activities at 34 DAP and 46 DAP due to problems with the assay of some enzymes. The activities of CL, AI, and SUS were significantly increased, consistent with the up-regulated expression of *CWINV4*, *BraA03g023380.3C*, *CEL1*, *BGLU16,* and *SUS3* (Figure 7 and Figure 8B,D,E). Similarly, the down-regulation of *BraA06g003260.3C*, *BraA08g002960.3C*, *BraA05g019040.3C*, *BraA05g027230.3C*, *BAM1*, *BAM3*, *BAM3-like*, *BAM5*, *CHS1,* and *CHS3* resulted in a significant decrease in the activities of HK, β-amylase (BMY), and chalcone synthase (CHS) (Figure 7). The activity of α-amylase (AMY) and SPS at 46 DAP was close to that at 34 DAP, and DEGs encoding these two enzymes also did not appear in our transcriptome data (Figure 5B and Figure 8C,G). In general, the significant increase in AI and CL activities promoted sugar biosynthesis, while the significant decrease in BMY, HK, and CHS activities suppressed sugar loss.

## 3. Discussion

Sugar regulatory pathways are vital for metabolism during vegetable and fruit development and maturation [33]. The sweetness of vegetables and fruit depends mainly on the type and composition of sugars, which play key roles in flavor [10,48]. Sweetness, as an important indicator of wucai quality, increased significantly during the sugar maturation process. As research on sugar accumulation in wucai is limited, the sugar composition, sugar changes, and expression of genes related to sugar accumulation were analyzed during sugar transformation in wucai ”W16-19-5” herein.

As previously reported in Chinese cabbage, tomato, pumpkin, watermelon, and melon, a significant increase in soluble sugars occurred during ripening [28,34,42,49,50,51]. In our study, the change in soluble sugar in wucai was similar to those in the above fruit and vegetables during the maturation process. In addition, we found that the soluble sugar content at 28 DAP was relatively low compared to at 22 DAP (Figure 1A). Wucai is grown in autumn and winter, and the air temperature gradually decreases after sowing. The growth environment of wucai was simulated in a growth chamber herein, and lowering of the temperature was first initiated at 28 DAP. Thus, we inferred that the soluble sugar decreased at 28 DAP due to the change in temperature. The growth environment of wheat is similar to that of wucai, and D-galactose accumulated greatly at the late stage of development in wheat [52]. D-galactose, in addition to sucrose, glucose, and fructose, in wucai was measured at 34 DAP and 46 DAP. We found that sucrose did not increase significantly, whereas glucose and D-galactose did more than fructose.

Compared to 34 DAP, the ratio of glucose/soluble sugar increased significantly at 46 DAP (Figure 1F). Though there were no differences between the ratio of D-galactose/soluble sugar at the two periods, a great increase in their content occurred (Figure 1B,G). Carbohydrates also mainly constitute the differential metabolites during the ripening of kiwifruit and watermelon, which is consistent with our results [53,54]. In grape berry, sorghum stem, saffron corm, and melon, metabolites related to sugar accumulation were mainly enriched in fructose and mannose metabolism, starch and sucrose metabolism, glycolysis/gluconeogenesis, and pentose phosphate pathways [51,55,56,57]. We found that in addition to the pathways described above, galactose metabolism was also a significantly enriched pathway (Figure 3). The results showed that D-galactose and β-D-glucose were indeed the major accumulated sugars during the sugar transformation process and played a critical role in sugar accumulation.

Sweetness, one of the major traits of wucai, is a significant factor influencing wucai quality and is also an indicator of consumer preference [3]. In this study, we found that D-galactose and β-D-glucose, which have a sweet taste, were the major sugars in the sugar accumulation process in wucai (Table 1 and Figure 4). Therefore, the mechanism of accumulation of the two major sugar was analyzed using transcriptomics.

AI promoted the hydrolysis of not only sucrose, but also raffinose and stachyose [26]. CWINV and VINV activities were positively regulated by their encoding genes and they all were the AI [26,27,28]. The downregulation of *BFRUCT3* showed that sugar accumulation did not depend on the hydrolysis of sucrose in the vacuoles during wucai ripening. Thus, the up-regulation of *CWINV4* during the ripening of wucai resulted in a significant increase in AI activity, allowing for more D-galactose and β-D-glucose production. Wucai leaf is both the source tissue and the sink tissue. We found that the expression of *CWINV4* was significantly increased in wucai leaf compared to the other tissues at 46 DAP (Appendix A). This result was contrary to that of Chinese cabbage [5,42]. The IM is the main tissue of sugar accumulation in Chinese cabbage. *CWIN1* (CWINV), *NIN-like* (CINV), and *VIN4b* (VINV) had relatively lower expressions in the inner leading leaves than the external leading leaves during Chinese cabbage ripening, especially in IM [5]. Three INV genes (encoding β-fructofuranosidase 1, β-fructofuranosidase 6, and β-fructofuranosidase 3) were also significantly downregulated in the inner leaves of yellow-head Chinese cabbage [42]. In addition, the basic leucine zipper (bZIP) transcription factor (TF) GmbZIP123 promoted the expression of three CWINV genes (*CWINV1*, *CWINV3,* and *CWINV6*) by directly binding to their promoters, resulting in higher levels of glucose, fructose, and sucrose in soybean [58]. A pitaya WRKY TF *HpWRKY3* was associated with fruit sugar accumulation via the activation of the sucrose metabolic gene *HpINV2* [59]. While there was no bZIP TF detected herein, WRKY TFs were detected in this study. Identifying which WRKY TFs can work with *CWINV4* needs further analysis and verification.

The SPS activity did not change during the maturation of wucai, but SUS activity increased remarkably. In addition, one DEG (SUS3) encoding SUS was up-regulated in the transcriptome data, and no SPS DEGs were found, consistent with the enzyme activities (Figure 8B,C and Figure 5B). Therefore, it was inferred that *SUS3* promoted the synthesis of sucrose to offset the hydrolysis of sucrose under *CWINV4*. Starch degradation during ripening is a key additional process for D-glucose accumulation in fruit and is catalyzed by the action of amylases [60]. The activity of AMY and DPE increased during mango ripening with a concomitant decrease in the starch content of the fruit [13]. BMY activity and *BAMs* (*BAM1*, *BAM3*, *BAM3-like*, and *BAM5*) were significantly down-regulated (Figure 5B). *DPE* catalyzing starch conversion into D-glucose was also found to be downregulated (Figure 5B). However, there was no differential accumulation of starch during wucai ripening, due to the downregulation of *SS1* and *SBE3* for starch synthesis. It follows that the accumulation of β-D-glucose did not originate from starch degradation during wucai ripening.

The cellulose hydrolytic enzyme beta-1, 4-endoglucanase (E1) gene, from the thermophilic bacterium *Acidothermus cellulolyticus*, was overexpressed in rice through *Agrobacterium*-mediated transformation [61]. Hydrolysis of transgenic rice straw yielded 43% more reducing sugars than wild-type rice straw did [61]. It was found that overexpression of *EG* promoted the hydrolysis of cellulose, which is consistent with our study. Additionally, the up-regulated expression of BGL genes in a ripe rich-sugar mango variety showed that the genes could promote the accumulation of sugar [13]. There were no CBH DEGs detected in our transcriptome data (Figure 5B). However, we observed a significant increase in CL activity. It was inferred that *CEL1* and *BraA03g023380.3C* combined with *BGLU16* catalyzed cellulose into β-D-glucose. A β-glucosidase from *Clostridium cellulovorans* (CcBG) was fused with cellulosomal endoglucanase CelD (CtCD) from *Clostridium thermocellum* [62]. CtCD CcBG showed favorable specific activities on phosphoric-acid-swollen cellulose (PASC), with greater glucose production (2-fold) when compared with a mixture of the single enzymes, further supporting our conclusions [62]. The transcription levels in mature Chinese cabbage and rich-sugar mango were significantly higher than those of unmatured Chinese cabbage and low-sugar mango, which proved that the downregulated expression of *HK* led to the accumulation of more glucose [13,41]. Significantly reduced HK activity during maturation of wucai was accompanied by the down-regulated expression of HK DEGs (*BraA06g003260.3C*, *BraA08g002960.3C*, *BraA05g019040.3C*, and *BraA05g027230.3C*), which reduced the loss of D-glucose and led to more conversion of D-glucose to β-D -glucose. Similarly, the downregulation of HK activity reduced the phosphorylation of β-D-glucose, thereby promoting sugar accumulation.

We screened 26 DEGs possibly related to D-glucose and β-D-glucose accumulation by calculating the correlation between TOP100 DEGs in transcriptome and target metabolites. Interferon-related developmental regulator (IFRD) was mainly involved in plant salt tolerance, cold tolerance, and the ABA signal transduction pathway in previous reports [63,64,65]. As wucai gradually matured, the relative expression levels of *BraA09g036850.3C* were higher than those at 34 DAP, suggesting that the high expression of the gene during this process may promote sugar accumulation (Figure 7). Some scholars have pointed out the beneficial role of inositol in promoting sugar accumulation [66]. In the biosynthesis of inositol, the rate-limiting step is catalyzed by inositol-3-phosphate synthase (ISYNA) [67]. Thus, *BraA01g000700.3C* was speculated to be highly expressed after maturation to enhance sugar accumulation (Figure 7). S-adenosylhomocysteine hydrolase (SAHH) is a widespread enzyme in cells. Over-expression of *SlSAHH2* could enhance SAHH enzymatic activity in tomato development and ripening stages and resulted in a major phenotypic change of reduced ripening time from anthesis to breaker [68]. Interestingly, SAHH enzyme activity levels and *SlSAHH2* transcript levels appeared to be inconsistent in some tissues. For example, *SlSAHH2* was not significantly elevated in transgenic fruit, but its enzymatic activity remained at a high level [68]. From the above, it was assumed that *SAHH2* decreased during the ripening process, but it still maintained a high level of enzyme activity to promote ripening and sugar accumulation in wucai.

Sugars can be used as precursors and information-regulating molecules for synthesis of anthocyanins [69]. *CHI*, *CHS*, *FLS*, *F3H* (flavanone-3-hydroxylase), *PAL* (phenylalaninammo-nialyase), and *OMT1* that affect the synthesis and accumulation of anthocyanin were regulated by sugar [69,70]. For example, the expression of the petunia CHS gene in transgenic Arabidopsis leaves was induced by sugars [71]. *CHI*, *CHS*, *FLS*, and *OMT1* in wucai were down-regulated during ripening, where the measured CHS activity was also significantly decreased (Figure 7 and Figure 8). Different sugar sensing mechanisms exist in plants and respond to different sugars [72]. We speculate that in wucai, D-galactose and β-D-Glucose could have a negative effect on the synthesis of anthocyanin, and the down-regulation of *CHI*, *CHS*, *FLS*, and *OMT1* reduced the loss of anthocyanin synthetic precursors. There were 18 other DEGs that had a significant correlation with D-galactose, and they were all negatively correlated (Figure 6B). However, how these genes regulate sugar accumulation remains unknown, which needs the support of further studies.

This study is the first to report on sugar accumulation during the maturation process of wucai. We found that D-galactose and β-D-glucose were mainly accumulated during wucai ripening and are essential for improving the taste quality of the fruit. The upregulated expression of *CWINV4*, *CEL1*, *BGLU16,* and *BraA03g023380.3C* and downregulated expression of *BraA06g003260.3C, BraA08g002960.3C, BraA05g019040.3C,* and *BraA05g027230.3C* in the pathway might contribute to the accumulation of D-galactose and β-D-glucose. Twenty-six DEGs significantly related to D-galactose and β-D-glucose may regulate their accumulation in wucai. This research could support the quality grading of wucai and the breeding of excellent wucai lines.

## 4. Materials and Methods

### 4.1. Plant Materials and Growth Conditions

W16-19-5, a typical wucai cultivar line, was used in this study. This experiment was carried out at the breeding base of Anhui Agricultural University (Hefei, China). The seeds of the experimental variety were obtained from the Vegetable Genetics and Breeding Laboratory of Anhui Agricultural University. Seeds were sown in plugs in a greenhouse, and seedlings with 6–7 leaves were transplanted into pots containing a substrate and vermiculite at a volume ratio of 2:1. Subsequently, the seedings were grown in a growth chamber (0 DAP) at 25 ± 1 °C (day) and 15 ± 1 °C (night) with a 300 μmol·m^−2^·s^−1^ photon flux density and 70% relative humidity under a 16/8 h (day/night) photoperiod. At 28 DAP, the growth chamber was modified to 10 °C (day) and 4 °C (night), and the other conditions remained the same. The fourth and fifth fully expanded young leaves from the center of the plants, petiole, root, and stem were sampled. Fresh leaves were placed at 105 °C for 20 min and then dried at 75 °C for 24 h to obtain a dry sample. The first sampling was performed at 4 DAP and then at 5-day intervals, with sampling ending at 52 DAP (Appendix A). Fresh samples were immediately frozen in liquid nitrogen and maintained at −80 °C for analyses.

### 4.2. Measurement of Sugar Content and Enzyme Activity

Measurements of soluble sugar were carried out at nine sampling periods, namely 4 DAP, 10 DAP, 16 DAP, 22 DAP, 28 DAP, 34 DAP, 40 DAP, 46 DAP, and 52 DAP. The soluble sugar was measured according to the anthrone colorimetric method with slight modifications [73]. Fresh leaves (0.2 g) were boiled in ddH2O (10 mL) for 30 min and then filtered and homogenized (25 mL). The extract (0.5 mL) was added to 1.5 mL of ddH_2_O, 0.5 mL of anthrone ethyl acetate, and 5 mL of pure sulfuric acid. The absorbance was measured at 630 nm by a UV-vis spectrophotometer (TU1950, PERSEE).

Soluble sugar, sucrose, and fructose in the dry sample were measured at 34 DAP and 46 DAP by the anthrone colorimetric method with slight modifications [73]. Dried leaves (50 mg) were mixed with 4 mL of alcohol (80%, *v/v*) and shaken at 80 °C for 30 min. The residue was extracted with 80% alcohol. The two mixtures were configured to determine the described sugar content. The first mixture contained 0.25 mL of extract, 0.25 mL of ddH_2_O, and 50 µL of NaOH (2 mol/L) and was boiled at 90 °C for 5 min. The second mixture of 0.5 mL of extract and 2.5 mL of anthrone was boiled at 40 °C for 10 min. The corresponding absorbance values were measured at 620 nm. Glucose was extracted using a Solarbio reagent kit (Cat #BC1580; Beijing Solarbio Science & Technology Co., Ltd., Beijing, China). D-galactose was quantified using a kit (ADS-W-TDX046; Shanghai Kexing Trading Co., Ltd., Shanghai, China).

AI, CL, AMY, BMY, HK, SPS, and CHS activities were measured at 34 DAP and 46 DAP according to kits (Cat #BC0560, Cat #BC2540, Cat #BC2040, Cat #BC0740, Cat #BC0600, and Cat #BC0580; Beijing Solarbio Science & Technology Co., Ltd., Beijing, China. Cat # ml092866; Shanghai Enzyme-linked Biotechnology Co., Ltd., Shanghai, China), respectively.

### 4.3. Metabolomic Analysis

The extraction, detection, and quantitative analysis of metabolites in the samples were performed by Shanghai Lu-Ming Biotech Co., Ltd. (Shanghai, China) (https://www.lumingbio.com/, accessed on 3 January 2021). In brief, freeze-dried wucai leaf samples (80 mg) were weighed and extracted overnight at −20 °C with 20 μL of 2-chloro-l-phenylalanine (0.3 mg/mL) dissolved in methanol as an internal standard and 1 mL of mixture of methanol and water (7/3, *v/v*). The samples were centrifuged at 13,000 rpm and 4 °C for 15 min. The supernatants (150 μL) were collected and then filtered through 0.22 μm microfilters and transferred to LC vials. Sample extracts were filtered and analyzed by LC-MS/MS. All metabolites were identified by Progenesis QI (Waters Corporation, Milford, CT, USA) Data Processing Software, based on public databases (http://www.hmdb.ca/; http://www.lipidmaps.org/, accessed on 3 January 2021) and self-built databases. The GC-MS/MS analysis was similar to that of the LC-MS/MS analysis. Sixty milligrams of freeze-dried wucai leaves samples was weighed and combined with 40 μL of 2-chloro-l-phenylalanine (0.3 mg/mL) dissolved in methanol as an internal standard and 360 μL of cold methanol. Two milliliters of chloroform and 4 mL of water were added to the sample, which was ground and then extracted. The supernatant (200 μL) was transferred to a glass sampling vial for vacuum-drying at room temperature. Eighty microliters of 15 mg/mL of methoxyamine hydrochloride in pyridine was subsequently added, following which 80 μL of BSTFA (with 1% TMCS) and 20 μL of n-hexane were added into the mixture after rotating for 2 min and incubating at 37 °C for 90 min, which was then followed by vigorous vortexing for 2 min and then derivatization at 70 °C for 60 min. After 30 min at room temperature, the sample extracts were filtered and analyzed by GC-MS/MS. Metabolites were annotated through the LUG database (Untarget database of GC-MS/MS from Lumingbio). Metabolic alterations among experimental groups were visualized by principal component analysis (PCA) and (orthogonal) partial least-squares-discriminant analysis (O)PLS-DA. Group discrimination was ascertained based on VIP scores >1 obtained from the OPLS-DA model. Metabolites with VIP > 1 and *p*-value < 0.05 were considered differential metabolites.

The OPLS-DA S-plot was obtained from the OPLS-DA, with minor modification. All points representing DAMs in the figure are distributed in the first and third quadrants, similar to an S-shape, which is called an OPLS-DA S-plot. Metabolites that are significantly different are distributed in the upper left corner and lower right corner. Biomarker analysis was performed by MetaboAnalyst 5.0 (https://www.metaboanalyst.ca/,accessed on 27 August 2021).

### 4.4. RNA-Seq Analysis

The total RNA of the wucai leaf samples at the two sampling periods (34 DAP and 46 DAP) was extracted using a mirVana miRNA Isolation Kit (Ambion) according to the manufacturer’s instructions. The RNA integrity was evaluated using an Agilent 2100 Bioanalyzer (Agilent Technologies, Santa Clara, CA, USA). The samples with RNA Integrity Number (RIN) ≥ 7 were subjected to subsequent analysis. The libraries were constructed using a TruSeq Stranded mRNA LTSample Prep Kit (Illumina, San Diego, CA, USA) following the manufacturer’s instructions. Then, six cDNA libraries were sequenced on the Illumina sequencing platform (HiSeqTM 2500 or Illumina HiSeq X-Ten). Raw data (raw reads) were first processed using Trimmomatic [74], and then the low-quality reads were removed to obtain the clean reads for subsequent analyses. The clean reads were mapped to the B. rapa reference genome using HISAT2 [75]. Fragments Per Kilobase of transcript per Million mapped reads (FPKM) values and the read counts of each gene were obtained, respectively, by Cufflinks and HTSeqcount [76]. Differentially expressed unigenes (DEGs) were identified using the DESeq (2012) function estimateSizeFactors and nbinomTest, and q < 0.05 and |log_2_(fold change)| > 1 were set as the threshold for significant differential expression. KEGG pathway enrichment analysis of DEGs was performed in R software based on the hypergeometric distribution.

### 4.5. qRT-PCR Analysis

Twenty-eight genes were selected for qRT-PCR analysis, and a gene encoding actin was used as the internal reference gene. The total RNA of the wucai leaves was extracted using an RNA kit (Takara Biomedical Technology Co., Beijing, China). The primers designed by Primer software v6.0 (Premier Biosoft International, Palo Alto, CA, USA) are listed in Appendix A. The qRT-PCR was performed using the Hieff^®^ qPCR SYBR^®^ Green Master Mix (No Rox) (Yeasen, Shanghai, China). The relative mRNA expression level of genes was calculated using the 2^-ΔΔCT^ method [77].

### 4.6. Statistical Analysis

All data were analyzed using Origin 2020 64 Bit, Adobe Illustrator 2019, Excel 2019, Adobe Photoshop 2021, Cytoscape_v3.8.2, and SPSS 26.0 and were expressed as mean ± SD. Tukey’s post hoc test was used for mean comparisons using *p* < 0.05. All data were from three biological replications.

## 5. Conclusions

In the present study, LC-MS/MS, GC-MS/MS, and RNA-Seq profiling were performed to explore the molecular regulatory mechanisms of sugar accumulation during the maturity process of wucai. In the comparison of 46 DAP vs. 34 DAP, the number of DAMs associated with carbohydrates was prominent in LC-MS/MS and GC-MS/MS. The main ways of sugar accumulation were the pentose phosphate pathway, galactose metabolism, glycolysis/gluconeogenesis, starch and sucrose metabolism, and fructose and mannose metabolism in metabolome profiling. D-galactose and β-D-glucose, the two significantly accumulated metabolites, were identified as the main sugar to improving the taste quality of wucai during sugar transformation. Combined with the transcriptome data, the pathway of sugar accumulation and the interaction network of DEGs and the two sugars were generated. *CWINV4*, *CEL1*, *BGLU16*, and *BraA03g023380.3C*, which directly regulate sugar production, were significantly upregulated, and the enzymes activities (AI and CL) they encode showed the same results. Likewise, the expressions of HK (*BraA06g003260.3C*, *BraA08g002960.3C*, *BraA05g019040.3C*, and *BraA05g027230.3C*) and HK activity were both significantly decreased, reducing the metabolic loss of sugar. The 26 DEGs in the interaction network may regulate sugar accumulation through some unknown pathways. Among them, *BraA09g036850.3C*, *BraA01g000700.3C*, *SAHH2*, *CHI*, *CHS1*, *CHS3*, *FLS1*, and *OMT1* all have effects on D-galactose and β-D-glucose metabolism. These findings could help us understand the main substances and molecular regulation mechanism during the process of sugar accumulation.

## Figures and Tables

**Figure 1 ijms-24-04816-f001:**
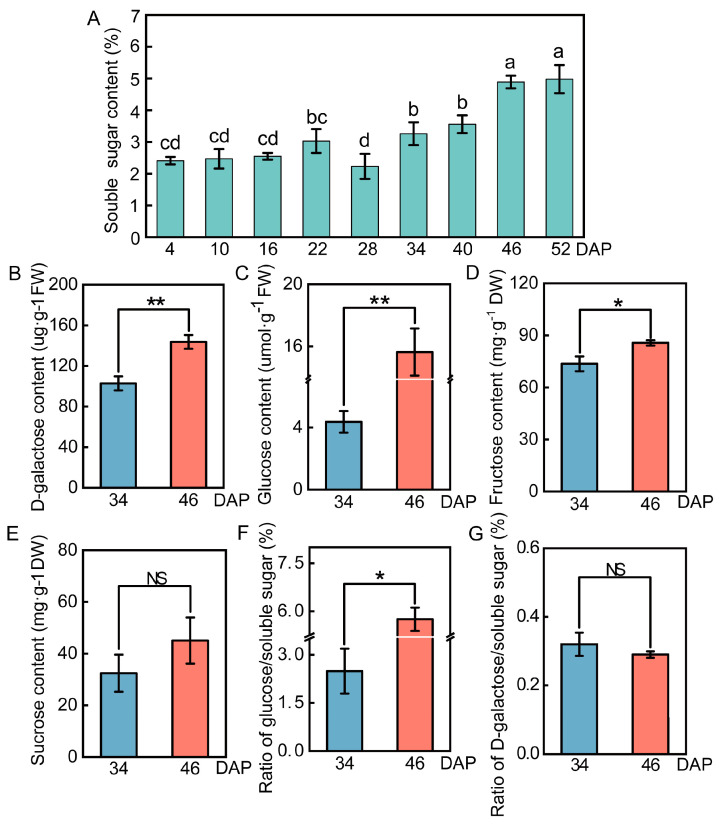
Sugar content in wucai leaves. (**A**) Soluble sugar content (%). (**B**) D-galactose content (μg·g^−1^ FW). (**C**) Glucose content (μmol·g^−1^ FW). (**D**) Fructose content (mg·g^−1^ DW). (**E**) Sucrose content (mg·g^−1^ DW). (**F**) Ratio of glucose/soluble sugar. (**G**) Ratio of D-galactose in/soluble sugar. FW and DW meant fresh weight and dry weight, respectively. Values presented are the mean ± SE (n ≥ 3), and bars with different letters represent significant differences at *p* < 0.05. * and ** meant *p* < 0.05 and *p* < 0.01, respectively. NS indicated no significant difference.

**Figure 2 ijms-24-04816-f002:**
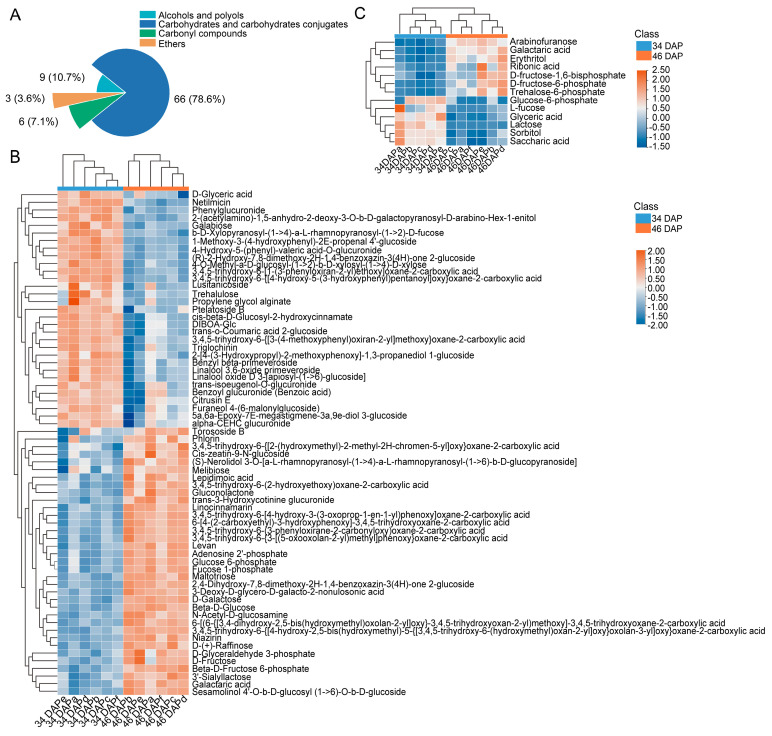
Classification and heatmap of DAMs in wucai leaves. (**A**) Classification of organooxygen compounds in each sub class category in LC-MS/MS. (**B**,**C**) Heatmap of DAMs as carbohydrates and carbohydrate conjugates in LC-MS/MS and GC-MS/MS. The data were derived from the numbers of DAMs in each sub-class category in the LC-MS/MS and GC-MS/MS analyses.

**Figure 3 ijms-24-04816-f003:**
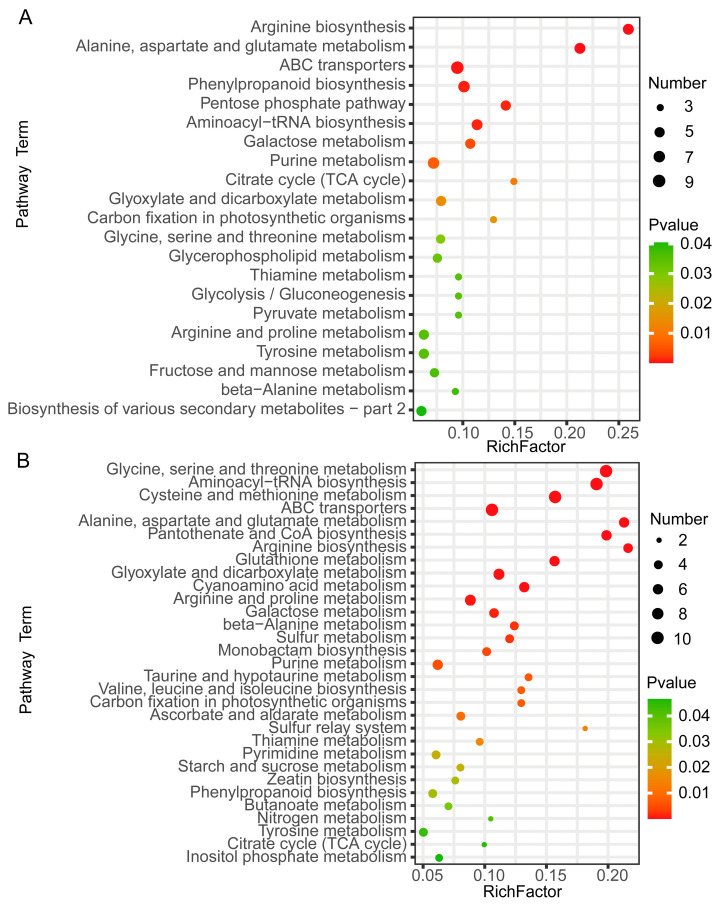
Enriched KEGG pathways with DAMs based on *p*-value < 0.05. (**A**) LC-MS/MS. (**B**) GC-MS/MS. The abscissa represents the rich factor: DAMs/total DAMs number. The size of the dot indicates the number of DAMs.

**Figure 4 ijms-24-04816-f004:**
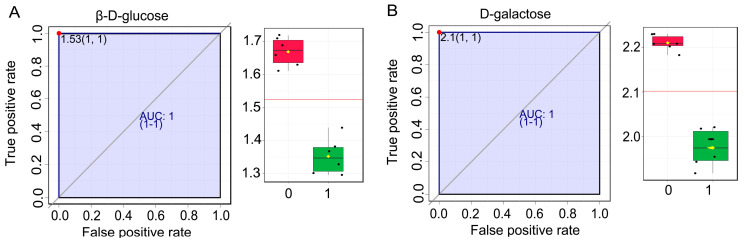
ROC and box plot of the biomarkers. (**A**) β-D-glucose. (**B**) D-galactose. The ROC curve closer to the left parietal corner indicates that the DAM has excellent sensitivity and specificity. The box plot intuitively shows the differences in expression abundance between the DAMs. The red box indicated 46 DAP, while the green one indicated 34 DAP.

**Figure 5 ijms-24-04816-f005:**
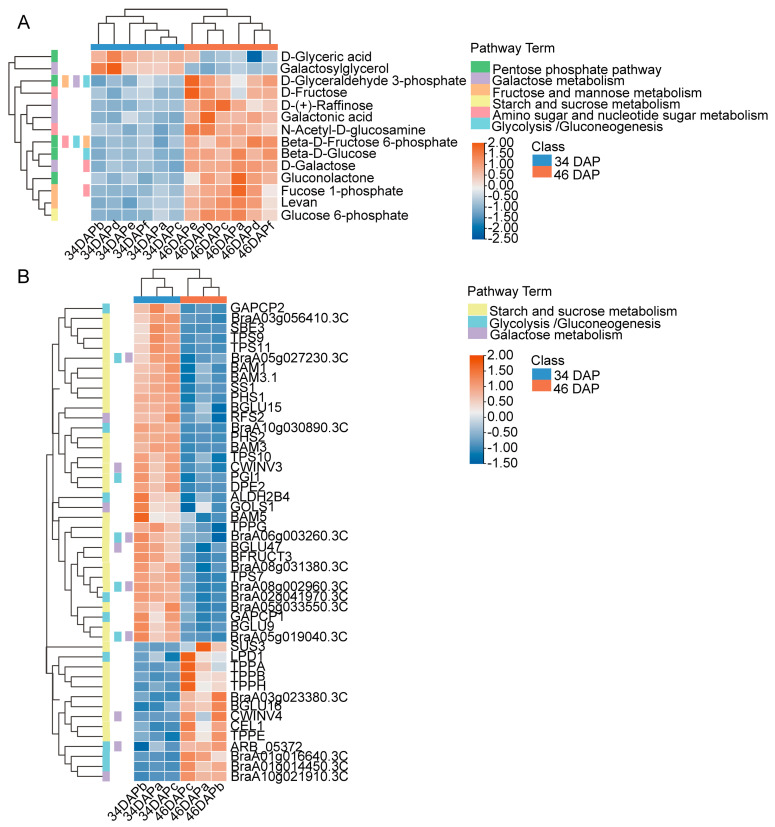
Heatmap of DAMs and DEGs in KEGG enrichment pathways in wucai leaves. (**A**) Heatmap of DAMs located in KEGG enrichment pathways in the LC-MS/MS analysis. (**B**) Heatmap of DEGs in galactose metabolism, glycolysis/gluconeogenesis, and starch and sucrose metabolism. The color lump on the left of the heatmap indicates the corresponding pathways. The heatmap was based on the expression abundance of the DAMs and Fragments Per Kilobase of transcript per Million mapped reads (FPKM) values of the DEGs.

**Figure 6 ijms-24-04816-f006:**
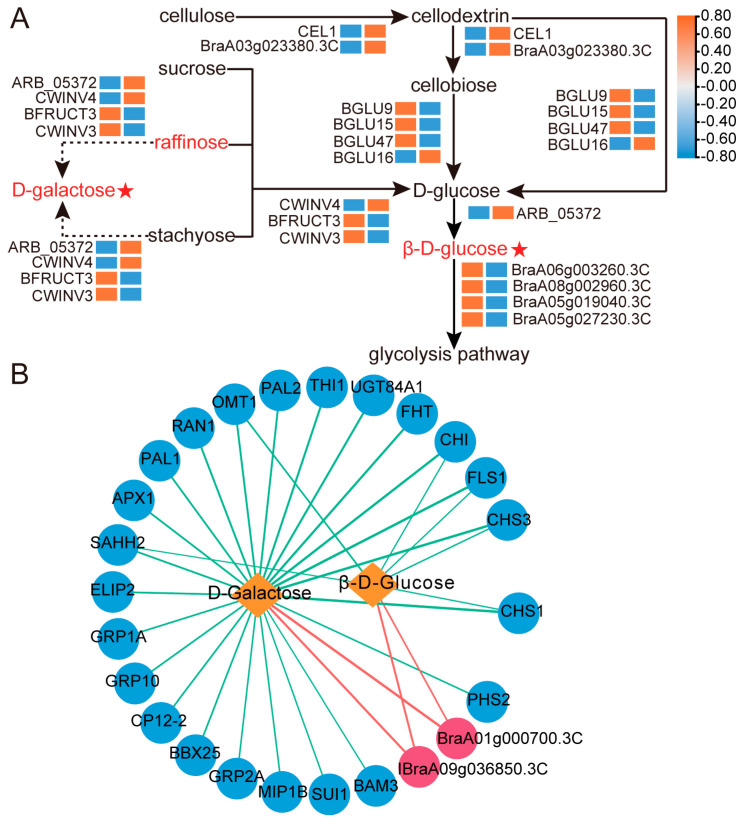
Accumulation pathway and Interact network of D-galactose and β-D-glucose. (**A**) Accumulation pathway. Metabolites upregulated or undifferentiated at 46 DAP are indicated in red and black. The red asterisk indicates a biomarker. The left and right columns of the heatmap represent 34 DAP and 46 DAP, respectively. Genes upregulated or downregulated are shown in orange and blue, respectively. The heatmap was based on the FPKM values of the DEGs. (**B**) Interact network. The circles with red and blue represent up-regulated and down-regulated DEGs, respectively. The orange diamond shape represents the two main accumulated sugars. The positive and negative correlations are represented by red and green lines, respectively. The thicker the line, the higher the correlation.

**Figure 7 ijms-24-04816-f007:**
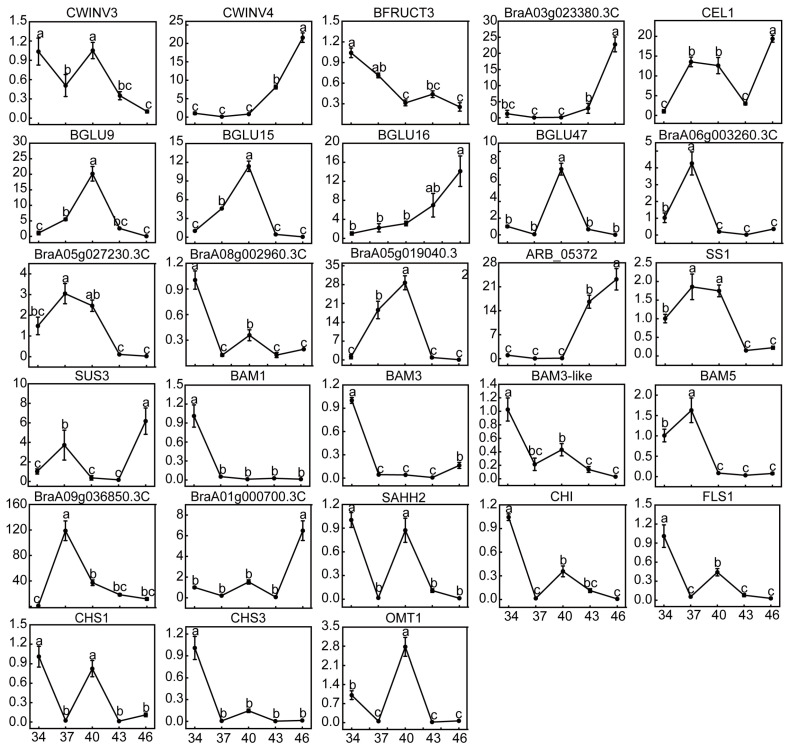
Relative expression level of DEGs during wucai ripening. The values obtained by the quantitative real-time PCR (qRT-PCR) represent the mean ± SE of three replicates. Bars with different letters are significantly different at *p* < 0.05.

**Figure 8 ijms-24-04816-f008:**
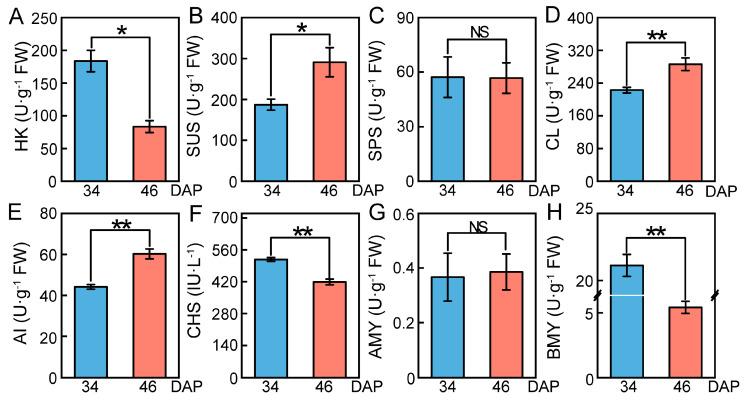
Enzyme activities in wucai leaves at 34 DAP and 46 DAP. (**A**) Hexokinase (U·g^−1^ FW). (**B**) Sucrose synthase (U·g^−1^ FW). (**C**) Sucrose phosphate synthase (U·g^−1^ FW). (**D**) Cellulase (U·g^−1^ FW). (**E**) Acid convertase (U·g^−1^ FW). (**F**) Chalcone synthase (IU·L^−1^). (**G**) α−amylase (U·g^−1^ FW). (**H**) β-amylase (U·g^−1^ FW). Values presented are the mean ± SE (n ≥ 3), * *p* < 0.05, ** *p* < 0.01 and NS indicated no significant difference between bars.

**Table 1 ijms-24-04816-t001:** Biomarker screening results by using MetaboAnalyst 5.0.

Metabolites	AUC	*T*-Test	Log_2_(FC)
3,4,5-trihydroxy-6-[4-hydroxy-3-(3-oxoprop-1-en-1-yl) phenoxy] oxane-2-carboxylic acid	1.0	3.9459 × 10^−6^	0.98885
D-Galactose	1.0	1.9657 × 10^−7^	0.94706
6-[4-(2-carboxyethyl)-3-hydroxyphenoxy]-3,4,5-trihydroxyoxane-2-carboxylic acid	1.0	3.0412 × 10^−5^	0.62342
Pheophorbide a	1.0	3.2816 × 10^−7^	3.142
Quinoline-3-carboxamides	1.0	6.0721 × 10^−7^	−2.6307
ibandronate	1.0	3.3144 × 10^−9^	1.3004
5′-Butyrylphosphouridine	1.0	2.1576 × 10^−6^	−1.0662
Phosphatidyl glycerol	1.0	2.0603 × 10^−9^	2.8388
Beta-D-Glucose	1.0	6.6146 × 10^−7^	1.2187
(s)C(S)S-S-Methylcysteine sulfoxide	1.0	2.6679 × 10^−7^	2.2726
3,4,5-trihydroxy-6-(3-phenyloxirane-2-carbonyloxy) oxane-2-carboxylic acid	1.0	5.1913 × 10^−6^	0.98688
Trehalulose	1.0	9.5148 × 10^−4^	−0.77498
Levan	1.0	3.2252 × 10^−5^	0.9449

## Data Availability

The data for RNA-sequencing are available at the National Center for Biotechnology Information (NCBI) with accession number PRJNA898258. The data for Metabonomics are available in the EMBL-EBI MetaboLights database with accession number MTBLS5097 and MTBLS5096.

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
