# Peer review of "Combined Metabolome and Transcriptome Analysis Elucidates Sugar Accumulation in Wucai (*Brassica campestris* L.)"

_ijms, 2023, doi:10.3390/ijms24054816_

Round 1
Reviewer 1 Report
The main comments as follow:
1. In Abstract, authors should draw the clearer conclusion, like the potential main effective genes in sugar accumulation.
2. Introduction section is too simple, and suggest to supplement some reports on sugar and its accumulation in wucai.
3. Please explain the biomarker, and elucidate why you could determine the biomarker only through comparing the divergency of matobolosm and transcriptom data between various stages of wucai.
4. In Figure 1A, the second right number is miss, please add it.
5. In Result section, P5, L163-167, authors only described what they conducted, but no results.
6. In subsection of Results, 2.6. Coexpression analysis of genes related to D-galactose and β-D-glucose accumulation. Authors should try to explain which members might be the key factors in sugar accumulation, and how they play roles in sugar pathway, instead of simply descriping which members involved, and what they have changed in these pathways.
7. The title of subsection (2.7. Changes in enzyme activities and relative expression levels of DEGs) should be consistent in the orders of following contents, suggest the title should be revised as 2.7. Changes in relative expression levels and enzyme activities of DEGs.
8. Disscussion section should be re-written more logically, not spread out all contents.
9. In Reference section, please carefully check and correct all the reference list, like: the format of journal names should be uniform.
Author Response
Dear Editors and Reviewers:
I am very grateful for your letter. Many thanks for the editors and the reviewers for your valuable advice and comments on our manuscript entitled “Combined Metabolome and Transcriptome Analysis Elucidates Sugar Accumulation in Wucai (Brassica campestris L.).” (ijms-2104373). I carefully read the comments of the reviewers and editors, responded all the comments and corrections passed by the reviewers and the editors in our revised manuscript. The following are modifications and explanations of our manuscript (ijms-2104373). Revised portion are marked using “Track Changes” functions in the paper.
Thank you again for your reply and look forward to your next letter.
Best regards.
Sincerely yours,
Lingyun Yuan
Associate professor
Department of Horticulture, Anhui Agricultural University
130 Changjiang West Road, Hefei, Anhui, China
Tel/Fax: 0551-65786185; E-mail: ylyun99@163.com
Response to Reviewer 1 comments
Point 1. In Abstract, authors should draw the clearer conclusion, like the potential main effective genes in sugar accumulation.
Response: Thanks for your suggestion.
In this paper, CWINV4, CEL1, BGLU16, BraA03g023380.3C, BraA06g003260.3C, BraA08g002960.3C, BraA05g019040.3C, and BraA05g027230.3C had a direct regulatory effect on sugar accumulation in wucai through the analysis in subsection of Results 2.6 and discussion. The 26 DEGs in the interaction network may regulate sugar accumulation through some unknown pathways. Even if BraA09g036850.3C, BraA01g000700.3C, SAHH2, CHI, CHS1, CHS3, FLS1 and OMT1 among them all have effects on D-galactose and β-D-glucose metabolism.
Revised manuscript lines 25-28: revised to “CWINV4, CEL1, BGLU16, and BraA03g023380.3C had positive correlation with the accumulation of sugar accumulation in wucai. The lower expression of BraA06g003260.3C, BraA08g002960.3C, BraA05g019040.3C, and BraA05g027230.3C promoted sugar accumulation during ripening of wucai.”.
Point 2. Introduction section is too simple, and suggest to supplement some reports on sugar and its accumulation in wucai.
Response: Thanks for reviewing our paper and suggestion.
So far, there is no report on sugar accumulation in wucai. Therefore, we added reports on sugar accumulation related to Chinese cabbage. In addition, we added information on sugar metabolism-related enzymes and biomarkers. Please see the revised manuscript for a complete introduction.
Revised manuscript lines 61-112: added “Sugar accumulation comes mainly from the transport of photosynthetic products, sucrose being the form of transport in most plants, and a number of key enzymes can be involved in regulating sugar metabolism and thus the composition and content of sugars [16]. Sucrose phosphate synthase (SPS), one of the key enzymes in plant sucrose synthesis, catalyzes the production of sucrose as an irreversible reaction and is the rate-limiting enzyme for the synthesis of sucrose [17]. SPS activity is positively correlated with sucrose accumulation [18]. Transcript levels of SPS increased with sucrose accumulation during ripening in watermelon and banana [19,20]. In addition, the expression pattern of SPS in pineapple and potato all showed that its expression was related to sucrose metabolism [21,22]. Sucrose synthase (SUS) catalyzes both the breakdown of sucrose to UDP glucose and the synthesis of sucrose [17]. SUS is also one of the key enzymes for the entry of sucrose into various metabolic pathways, regulating the ability of the crop to metabolize sucrose and the amount of sucrose input [23]. During the development of apple fruit, with the accumulation of sucrose, the expression of MdSUSY2, MdSUSY3, and MdSUSY4 decreased obviously, indicating that SUS mainly played a major role in the decomposition of apple sucrose [24]. The expression of CitSus5 was increasing while that of CitSus6 was gradually decreasing during fruit development in citrus, suggesting that SUS was involved in reversible reactions in citrus, possibly both synthesizing and breaking down sucrose [25]. Invertase (INV), also called sucrase, can hydrolyze sucrose into glucose and fructose. According to the site where the enzyme is present on the cell, INV mainly consists of cell wall convertase (CWINV), vesicle convertase (VINV) and cytoplasmic convertase (CINV) [26]. Based on the optimum PH of the enzyme, CWINV and VINV can be classified as acid convertase (AI), while CINV is a neutral invertase (NI) [26]. Numerous studies showed that there is a significant negative correlation between the activity of INV and sucrose accumulation in fruits [12,22,26,27].. In tomato fruits, CWINV and VINV are encoded by LIN and VI, respectively. LIN5, LIN7, LIN8, LIN9, and VI were upregulated by silencing SWEET7 (Sugars Will Eventu-ally be Exported Transporters) and SWEET14 to increase CWINV and VINV activity [28]. It can be seen that the upregulation of CWINV and VINV increases the activities of AI, thereby promoting the hydrolysis of sucrose. CWINV is typically considered as a sink-specific enzyme, and its activity is usually low in source leaves [29]. However, both MdCWINVs (MdCWINV2 and MdCWINV3) identified in apple had lower expression levels in the fruit than in the leaves, and the transcript levels of MdCWINV2 and MdCWINV3 declined dramatically during maturation [24]. Hexokinase (HK) could catalyze the phosphorylation of hexose, which could catalyze the conversion of glucose into glucose -6- phosphate (glucose -6P), and then enter the glycolytic pathway [30]. Overexpression of the HK was able to cause a significant reduction in the sugar content of plants [31].
Besides INV, SPS, and SUS, another enzyme related to sugar accumulation and metabolism in watermelon fruits is α -galactosidase [32]. Stachyose and raffinose are the main transportation forms of photosynthetic products in Cucurbitaceae plants, which can be decomposed by α -galactosidase to produce sucrose and galactose [33,34]. cellulase (CL) is an important enzyme complex, mainly consisting of endoglucanase (EG), exoglucanase (CBH) and β-glucosidase (BGL), which hydrolyze cellulose to form glucose [35–37]. In previous studies, cellulose was considered to be related to the softening of crops during development [38,39]. Nevertheless, in biomass utilization, CL is employed to hydrolyze cellulose in multiple steps to generate glucose [40]. In studies of sugar accumulation in Chinese cabbage that is more closely related to wucai, it was noted that BraA01gHT4 and BraA03gHT7 were positively correlated with soluble sugar content (mainly fructose and glucose) of inner lobe, while BraA03gFRK1, BraA09gFRK3, BraA06gSPS2 and BraA03gHT3 were negatively correlated with sugar content [41]. Furthermore, the high expression of SUS1 was considered to promote the accumulation of fructose and glucose in leaf balls of Chinese cabbage [42].”.
Revised manuscript lines 121-129: added “A biomarker is a characteristic biochemical index, which can be objectively measured to provide information about the biological process of the organism [45]. Metabonomics pays attention to the changes of small-molecule metabolites in organisms, which provides the possibility for identifying objective biomarkers. Scholars established and analyzed OPLS-DA model or OPLS-DA-Splot map, and then potential biomarkers can be found in the project based on variable importance in the projection (VIP) score >1 [46,47]. ROC (receiver operating characteristic curve) and AUC (area under ROC curve) diagnostics were performed using the online software MetaboAnalyst to identify potential biomarkers [46,47].”.
Point 3. Please explain the biomarker, and elucidate why you could determine the biomarker only through comparing the divergency of metabolism and transcriptome data between various stages of wucai.
Response: Thanks for your suggestion and reminder.
A biomarker is a characteristic biochemical index, which can be objectively measured to provide information about the biological process of the organism. In this paper, biomarkers were identified based on metabolomic data with reference to previous methods. Scholars established and analyzed OPLS-DA model or OPLS-DA-S-plot map, and then potential biomarkers can be found in the project based on variable importance in the projection (VIP) score >1. ROC (receiver operating characteristic curve) and AUC (area under ROC curve) diagnostics were performed using the online software MetaboAnalyst to identify potential biomarkers. The closer the AUC is to 1, the better diagnostic performance is demonstrated. The Log2 fold change (FC) and VIP of D-galactose and β-D-glucose was greater indicating significant difference accumulation, and AUC of them was both 1, and thus were identified as biomarkers of sugar accumulation during the maturation of wucai.
Revised manuscript lines 121-129: added “A biomarker is a characteristic biochemical index, which can be objectively measured to provide information about the biological process of the organism [45]. Metabonomics pays attention to the changes of small-molecule metabolites in organisms, which provides the possibility for identifying objective biomarkers. Scholars established and analyzed OPLS-DA model or OPLS-DA-Splot map, and then potential biomarkers can be found in the project based on variable importance in the projection (VIP) score >1 [46,47]. ROC (receiver operating characteristic curve) and AUC (area under ROC curve) diagnostics were performed using the online software MetaboAnalyst to identify potential biomarkers [46,47].”.
Point 4. In Figure 1A, the second right number is miss, please add it.
Response: Thanks for your reminder.
“46” has been added to Figure 1A.
Point 5. In Result section, P5, L163-167, authors only described what they conducted, but no results.
Response: Thanks for your reminder and suggestion.
According to your suggestion, we then added the corresponding results.
Revised manuscript lines 236-239: The metabolites that accumulated significantly in these enrichment pathways are D-glycoldehyde3-phosphate, D-fructose, D-(+)-raffinose, Galactonic acid, N-acetyl-D-glucosamine, β-D-fructose 6-phosphate, β-D-Glucose, Gluconolactone, Fucose 1-phosphate, levan, and Glucose 6-phosphate.
Point 6. In subsection of Results, 2.6. Coexpression analysis of genes related to D-galactose and β-D-glucose accumulation. Authors should try to explain which members might be the key factors in sugar accumulation, and how they play roles in sugar pathway, instead of simply describing which members involved, and what they have changed in these pathways.
Response: Thanks for your reminder and suggestion.
We have made the modified in subsection of Results 2.6 as you suggested.
Revised manuscript lines 302-324: revised to “According to the major two sugars and related DEGs, we constructed an accumulation pathway of D-galactose and β-D-glucose (Figure 6A). In this way, there were three DEGs encoding INV, namely CWINVs (CWINV3, CWINV4) and VINV(BRFUCT3), all of which encode AI. Of these genes, only the expression of CWINV4 was up-regulated. Raffinose and stachyose located in galactose metabolic pathway were decomposed into D-galactose under AI (CWINV4). In the meantime, raffinose and stachyose were hydrolyzed into D-glucose under the action by the same gene. CWINV4 was also present in the starch and sucrose metabolic pathway, converting sucrose to D-glucose by hydrolysis. Moreover, cellulose in the starch and sucrose metabolic pathway was hydrolyzed to generated D-glucose. There were six DEGs associated with cellulose hydrolysis, EG (BraA03g023380.3C, CEL1) were up-regulated, while only one (BGLU16) of the BGL DEGs (BGLU16, BGLU9, BGLU15, and BGLU47) was up-regulated. Under the synergistic effect of BraA03g023380.3C, CEL1, and BGLU16, cellulose was gradually hydrolyzed into D-glucose. Aldose 1-epimerase (AEP) was able to catalyze the conversion of D-glucose to β-D-glucose. The generated D-glucose was converted to β-D-glucose by up-regulated expression of ARB_05372 (AEP). HK could phosphorylate β-D-glucose to β-D-Glucose 6-phosphate (β-D-glucose 6P), which later entered the glycolysis pathway. The four HK DEGs identified in this paper (BraA06g003260.3C, BraA08g002960.3C, BraA05g019040.3C, and BraA05g027230.3C) were all down-regulated, reducing the phosphorylation of β-D-glucose and promoting the accumulation of the sugar. The genes (galactokinase) catalyzing D-galactose were not differentially expressed, which showed that the accumulation of D-galactose mainly depended on AI under the action of CWINV4 during the maturation process of wucai.”.
Point 7. The title of subsection (2.7. Changes in enzyme activities and relative expression levels of DEGs) should be consistent in the orders of following contents, suggest the title should be revised as 2.7. Changes in relative expression levels and enzyme activities of DEGs.
Response: Thanks for your suggestion.
Revised manuscript lines 355: revised to “2.7. Changes in relative expression levels of DEGs and enzyme activities.”.
Point 8. Discussion section should be re-written more logically, not spread out all contents.
Response: Thanks for your reminder and suggestion.
In view of your proposal, while retaining some of the valid contents of the discussion, we have described the contents of the discussion more logically. Please see the revised manuscript for a complete discussion.
Revised manuscript lines 408-412: added to “The growth environment of wheat is similar to that of wucai, and D-galactose accumulated greatly at the late stage of development in wheat [52]. D-galactose in addition to sucrose, glucose, and fructose in wucai was measured at 34 DAP and 46 DAP. We found that sucrose did not increase significantly, whereas glucose and D-galactose did more than fructose.”
Revised manuscript lines 449-507: revised to “AI promoted the hydrolysis of not only sucrose, but also raffinose and stachyose [26]. CWINV and VINV activities were positively regulated by their encoding genes and they all were the AI [26–28]. The downregulation of BFRUCT3 showed that sugar accumulation did not depend on the hydrolysis of sucrose in the vacuoles during wucai ripening. Thus, the up-regulated of CWINV4 during the ripening of wucai resulted in a significant increase in AI activity, allowing for more D-galactose and β-D-glucose production. Wucai leaf is both the source tissue and the sink tissue. We found that the expression of CWINV4 was significantly increased in wucai leaf compared to the other tissues at 46 DAP (Supplementary Figure S7). This result was contrary to that of Chinese cabbage [5,42]. The IM is the main tissue of sugar accumulation in Chinese cabbage. CWIN1 (CWINV), NIN-like (CINV), and VIN4b (VINV) had relatively lower expression in the inner leading leaves than the external leading leaves during Chinese cabbage ripening, especially in IM [5]. Three INV genes (encoding β-fructofuranosidase 1, β-fructofuranosidase 6, and β-fructofuranosidase 3) were also significantly downregulated in the inner leaves of yellow-head Chinese cabbage [42]. In addition, the basic leucine zipper (bZIP) transcription factor (TF) GmbZIP123 promoted the expression of three CWINV genes (CWINV1, CWINV3, and CWINV6) by directly binding to their promoters, resulting in higher levels of glucose, fructose, and sucrose in soybean [58]. A pitaya WRKY TF HpWRKY3 was associated with fruit sugar accumulation via the activation of the sucrose metabolic gene HpINV2 [59]. While there was no bZIP TF detected herein, WRKY TFs were detected in this study. Identifying which WRKY TFs can work with CWINV4 needs further analysis and verification.
The SPS activity did not change during the maturation of wucai, but SUS activity increased remarkably. Also, one DEG (SUS3) encoding SUS was up-regulated in the transcriptome data, and no SPS DEGs was found, consistent with the enzyme activities (Figure 8 B-C and Figure 5B). Therefore, it was inferred that SUS3 promoted the synthesis of sucrose to offset the hydrolysis of sucrose under CWINV4. Starch degradation during ripening is a key additional process for D-glucose accumulation in fruit and is catalyzed by the action of amylases [60]. The activity of AMY and DPE increased during mango ripening with a concomitant decrease in the starch content of the fruit [13]. BMY activity and BAMs (BAM1, BAM3, BAM3-like, and BAM5) were significantly down regulated (Figure 5B). DPE catalyzing starch conversion into D-glucose was also found to be downregulated (Figure 5B). However, there was no differential accumulation of starch during wucai ripening due to the downregulation of SS1 and SBE3 for starch synthesis. It follows that the accumulation of β-D-glucose did not originate from starch degradation during wucai ripening.
The cellulose hydrolytic enzyme beta-1, 4-endoglucanase (E1) gene, from the thermophilic bacterium Acidothermus cellulolyticus, was overexpressed in rice through Agrobacterium-mediated transformation [61]. Hydrolysis of transgenic rice straw yielded 43% more reducing sugars than wild-type rice straw [61]. It was found that overexpression of EG promoted the hydrolysis of cellulose, which is consistent with our study. Additionally, the up-regulated expression of BGL genes in a ripe rich-sugar mango variety showed that the genes could promote the accumulation of sugar [13]. There were no CBH DEGs detected in our transcriptome data (Figure 5B). However, we observed a significant increase in CL activity. It was inferred that CEL1, BraA03g023380.3C combined with BGLU16 catalyzed cellulose into β-D-glucose. A β-glucosidase from Clostridium cellulovorans (CcBG) was fused with cellulosomal endoglucanase CelD (CtCD) from Clostridium thermocellum [62]. CtCD CcBG showed favorable specific activities on phosphoric acid-swollen cellulose (PASC), with greater glucose production (2-fold) when compared with a mixture of the single enzymes, further supporting our conclusions [62]. The transcription levels in mature Chinese cabbage and rich-sugar mango were significantly higher than that of unmatured Chinese cabbage and low-sugar mango, which proved that the downregulated expression of HK led to the accumulation of more glucose [13,41]. Significantly reduced HK activity during maturation of wucai was accompanied by down-regulated expression of HK DEGs (BraA06g003260.3C, BraA08g002960.3C, BraA05g019040.3C, and BraA05g027230.3C), which reduced the loss of D-glucose and lead to more conversion of D-glucose to β-D -glucose. Similarly, downregulation of HK activity reduced phosphorylation of β-D-glucose, thereby promoting sugar accumulation.”
Revised manuscript lines 619-624: revised “Interferon related developmental regulator (IFRD) was mainly involved in plant salt tolerance, cold tolerance, and ABA signal transduction pathway in previous reports [63–65].”.
Revised manuscript lines 625-638: revised to “Some scholars have pointed out beneficial role of inositol in promoting sugar accumulation [66]. In the biosynthesis of inositol, the rate-limiting step is catalyzed by inositol-3-phosphate synthase (ISYNA) [67]. So, BraA01g000700.3C was speculated to be highly expressed after maturation to enhance sugar accumulation (Figure 7). S-adenosylhomocysteine hydrolase (SAHH) is a widespread enzyme in cells. Over-expression of SlSAHH2 could enhanced SAHH enzymatic activity in tomato development and ripening stages and resulted in a major phenotypic change of reduced ripening time from anthesis to breaker [68]. Interestingly, SAHH enzyme activity levels and SlSAHH2 transcript levels appeared to be inconsistent in some tissues. For example, SlSAHH2 were not significantly elevated in transgenic fruit, but its enzymatic activity remained at a high level [68]. From the above, it was assumed that SAHH2 decreased during the ripening process, but it still maintained a high level of enzyme activity to promote ripening and sugar accumulation in wucai.”.
Revised manuscript lines 413-427, 433-436, 508-616, 621-623, 640-657, 665-667: The content of the original manuscript was deleted.
Point 9. In Reference section, please carefully check and correct all the reference list, like: the format of journal names should be uniform.
Response: Thanks for your reminder.
Due to the rearrangement of the introduction and discussion, the contents and numbers of the references have changed. Therefore, this document does not list the revisions of references. Please see the complete revised manuscript.
Reviewer 2 Report
Information on sugar accumulation in the leaves during maturation of wucai (Brassica campestris L. ssp. chinensis var. rosularis Tsen) is limited at present. This study employed LC-MS/MS, GC-MS/MS and RNA-Seq to figure out the molecular mechanism of sugar transformation in wucai during the maturation process. Results showed that D-galactose and β-D-glucose were identified as the major components of sugar accumulation in wucai, and several potential key genes were identified based on RNA-Seq analysis, which helps further studies focus on the biosynthesis pathway of D-galactose and β-D-glucose with deeper functional characterization of related genes.
The presented article “Combined Metabolome and Transcriptome Analysis Elucidates Sugar Accumulation in Wucai (Brassica campestris L.)” investigated metabolomic and transcriptomic changes between different developmental stages of Wucai (Brassica campestris L.), which is devoted to a pronounced applied value for Wucai quality. In my opinion, this manuscript could be acceptable for publishing in “International Journal of Molecular Sciences” with minor revisions.
1. The Supplementary Figure - “Pictures of wucai used in this study” should be referred after at nine sampling periods in Line 75.
2. “46” is missing in the x-axis of Figure 1A.
3. Line 114: “Supplementary Figure A2B” should be revised to “Supplementary Figure S2B”.
4. Line 526-527: there are redundant spaces and one enzyme activity name (α-amylase) is missing.
Author Response
Dear Editors and Reviewers:
I am very grateful for your letter. Many thanks for the editors and the reviewers for your valuable advice and comments on our manuscript entitled “Combined Metabolome and Transcriptome Analysis Elucidates Sugar Accumulation in Wucai (Brassica campestris L.).” (ijms-2104373). I carefully read the comments of the reviewers and editors, responded all the comments and corrections passed by the reviewers and the editors in our revised manuscript. The following are modifications and explanations of our manuscript (ijms-2104373). Revised portion are marked using “Track Changes” functions in the paper.
Thank you again for your reply and look forward to your next letter.
Best regards.
Sincerely yours,
Lingyun Yuan
Associate professor
Department of Horticulture, Anhui Agricultural University
130 Changjiang West Road, Hefei, Anhui, China
Tel/Fax: 0551-65786185; E-mail: ylyun99@163.com
Response to Reviewer 2 comments
Point 1. The Supplementary Figure - “Pictures of wucai used in this study” should be referred after at nine sampling periods in Line 75.
Response: Thanks for your suggestion.
In the unamended manuscript, "Pictures of wucai used in this study" was “Supplementary Figure S7”. If "Pictures of wucai used in this study" is referred after at nine sampling periods in Line 75, it needs to be modified to “Supplementary Figure S1”, and the order of other “Supplementary Figures” also needs to be revised.
Revised manuscript lines 139: “(Supplementary Figure S1)” was referred after at nine sampling periods in Line 146.
Point 2. “46” is missing in the x-axis of Figure 1A.
Response: Thanks for your reminder.
Figure 1A has been modified.
Point 3. Line 114: “Supplementary Figure A2B” should be revised to “Supplementary Figure S2B”.
Response: Thanks for your reminder.
Due to your first comment, the order of all the “Supplementary Figures” in the revised version has been modified. Therefore, "Supplementary Figure A2B" should be revised to "Supplementary Figure S3B".
Revised manuscript lines 181: "Supplementary Figure A2B" was revised to "Supplementary Figure S3B".
Point 4. Line 526-527: there are redundant spaces and one enzyme activity name (α-amylase) is missing.
Response: Thanks for your reminder.
AMY is the abbreviation of α-amylase. “α-amylase (AMY)” has been mentioned in line 381 of the revised manuscript, and “AMY” should be used in line 719.
Revised manuscript lines 719: redundant spaces have been removed and added “AMY”.
Reviewer 3 Report
The manuscript entitled “Combined Metabolome and Transcriptome Analysis Elucidates Sugar Accumulation in Wucai (Brassica campestris L.)” intended for publication in International Journal of Molecular Sciences is an interesting paper and relevant to special issue Sugar Transport, Metabolism and Signalling in Plant, however I think that manuscript needs some improvements.
Generally, the paper is relatively straightforward and well written, however some parts of manuscript need more attention. The Authors should improve Keywords (remove some title repetitions) and more specify the purpose of the study. I think, the Authors could modify the Introduction (too short) and Discussion (too long). I think that Introduction could be longer and more in detail - must contain basic information about the enzymes involved in sugar metabolism (some fragments can be transferred from the text of the Discussion). Descriptions/ captions of figures could be made in larger fonts, and some diagrams should be enlarged (e.g. Figs. 2, 3, 5, or Suppl. Fig.2). All abbreviations used should be explained, including those in figure captions. The Discussion chapter could have been shorter - without the basic information about the metabolism of carbohydrates (e.g. see lines: 298-308, p. 12, l 344- 363, 399-406). The Authors should check more carefully the Reference list, and improve it. In addition, there are small mistakes in the text of manuscript, including Reference list, that need to be corrected by Authors (e.g. lines: 22, 498, 703, 704, 768, 796, 846, 859, 862, 871, 877, 888, 898).
Author Response
Dear Editors and Reviewers:
I am very grateful for your letter. Many thanks for the editors and the reviewers for your valuable advice and comments on our manuscript entitled “Combined Metabolome and Transcriptome Analysis Elucidates Sugar Accumulation in Wucai (Brassica campestris L.).” (ijms-2104373). I carefully read the comments of the reviewers and editors, responded all the comments and corrections passed by the reviewers and the editors in our revised manuscript. The following are modifications and explanations of our manuscript (ijms-2104373). Revised portion are marked using “Track Changes” functions in the paper.
Thank you again for your reply and look forward to your next letter.
Best regards.
Sincerely yours,
Lingyun Yuan
Associate professor
Department of Horticulture, Anhui Agricultural University
130 Changjiang West Road, Hefei, Anhui, China
Tel/Fax: 0551-65786185; E-mail: ylyun99@163.com
Response to Reviewer 3 comments
Point 1. The Authors should improve Keywords (remove some title repetitions) and more specify the purpose of the study.
Response: Thanks for your suggestion.
Revised manuscript lines 35-36: revised to “wucai (Brassica campestris L.); D-galactose; β-D-glucose; sugar accumulation pathway; interact network.”.
Point 2. I think, the Authors could modify the Introduction (too short) and Discussion (too long). I think that Introduction could be longer and more in detail - must contain basic information about the enzymes involved in sugar metabolism (some fragments can be transferred from the text of the Discussion). The Discussion chapter could have been shorter - without the basic information about the metabolism of carbohydrates (e.g. see lines: 298-308, p. 12, l 344- 363, 399-406).
Response: Thanks for your reminder and suggestion.
According to your suggestion, we have made some modifications to the introduction. In view of your proposal, while retaining some of the valid contents of the discussion, we have described the contents of the discussion more logically. Please see the revised manuscript for a complete introduction and discussion.
Introduction:
Revised manuscript lines 61-112: added “Sugar accumulation comes mainly from the transport of photosynthetic products, sucrose being the form of transport in most plants, and a number of key enzymes can be involved in regulating sugar metabolism and thus the composition and content of sugars [16]. Sucrose phosphate synthase (SPS), one of the key enzymes in plant sucrose synthesis, catalyzes the production of sucrose as an irreversible reaction and is the rate-limiting enzyme for the synthesis of sucrose [17]. SPS activity is positively correlated with sucrose accumulation [18]. Transcript levels of SPS increased with sucrose accumulation during ripening in watermelon and banana [19,20]. In addition, the expression pattern of SPS in pineapple and potato all showed that its expression was related to sucrose metabolism [21,22]. Sucrose synthase (SUS) catalyzes both the breakdown of sucrose to UDP glucose and the synthesis of sucrose [17]. SUS is also one of the key enzymes for the entry of sucrose into various metabolic pathways, regulating the ability of the crop to metabolize sucrose and the amount of sucrose input [23]. During the development of apple fruit, with the accumulation of sucrose, the expression of MdSUSY2, MdSUSY3, and MdSUSY4 decreased obviously, indicating that SUS mainly played a major role in the decomposition of apple sucrose [24]. The expression of CitSus5 was increasing while that of CitSus6 was gradually decreasing during fruit development in citrus, suggesting that SUS was involved in reversible reactions in citrus, possibly both synthesizing and breaking down sucrose [25]. Invertase (INV), also called sucrase, can hydrolyze sucrose into glucose and fructose. According to the site where the enzyme is present on the cell, INV mainly consists of cell wall convertase (CWINV), vesicle convertase (VINV) and cytoplasmic convertase (CINV) [26]. Based on the optimum PH of the enzyme, CWINV and VINV can be classified as acid convertase (AI), while CINV is a neutral invertase (NI) [26]. Numerous studies showed that there is a significant negative correlation between the activity of INV and sucrose accumulation in fruits [12,22,26,27].. In tomato fruits, CWINV and VINV are encoded by LIN and VI, respectively. LIN5, LIN7, LIN8, LIN9, and VI were upregulated by silencing SWEET7 (Sugars Will Eventu-ally be Exported Transporters) and SWEET14 to increase CWINV and VINV activity [28]. It can be seen that the upregulation of CWINV and VINV increases the activities of AI, thereby promoting the hydrolysis of sucrose. CWINV is typically considered as a sink-specific enzyme, and its activity is usually low in source leaves [29]. However, both MdCWINVs (MdCWINV2 and MdCWINV3) identified in apple had lower expression levels in the fruit than in the leaves, and the transcript levels of MdCWINV2 and MdCWINV3 declined dramatically during maturation [24]. Hexokinase (HK) could catalyze the phosphorylation of hexose, which could catalyze the conversion of glucose into glucose -6- phosphate (glucose -6P), and then enter the glycolytic pathway [30]. Overexpression of the HK was able to cause a significant reduction in the sugar content of plants [31].
Besides INV, SPS, and SUS, another enzyme related to sugar accumulation and metabolism in watermelon fruits is α -galactosidase [32]. Stachyose and raffinose are the main transportation forms of photosynthetic products in Cucurbitaceae plants, which can be decomposed by α -galactosidase to produce sucrose and galactose [33,34]. cellulase (CL) is an important enzyme complex, mainly consisting of endoglucanase (EG), exoglucanase (CBH) and β-glucosidase (BGL), which hydrolyze cellulose to form glucose [35–37]. In previous studies, cellulose was considered to be related to the softening of crops during development [38,39]. Nevertheless, in biomass utilization, CL is employed to hydrolyze cellulose in multiple steps to generate glucose [40]. In studies of sugar accumulation in Chinese cabbage that is more closely related to wucai, it was noted that BraA01gHT4 and BraA03gHT7 were positively correlated with soluble sugar content (mainly fructose and glucose) of inner lobe, while BraA03gFRK1, BraA09gFRK3, BraA06gSPS2 and BraA03gHT3 were negatively correlated with sugar content [41]. Furthermore, the high expression of SUS1 was considered to promote the accumulation of fructose and glucose in leaf balls of Chinese cabbage [42].”.
Revised manuscript lines 121-129: added “A biomarker is a characteristic biochemical index, which can be objectively measured to provide information about the biological process of the organism [45]. Metabonomics pays attention to the changes of small-molecule metabolites in organisms, which provides the possibility for identifying objective biomarkers. Scholars established and analyzed OPLS-DA model or OPLS-DA-Splot map, and then potential biomarkers can be found in the project based on variable importance in the projection (VIP) score >1 [46,47]. ROC (receiver operating characteristic curve) and AUC (area under ROC curve) diagnostics were performed using the online software MetaboAnalyst to identify potential biomarkers [46,47].”.
Discussion:
Revised manuscript lines 408-412: added to “The growth environment of wheat is similar to that of wucai, and D-galactose accumulated greatly at the late stage of development in wheat [52]. D-galactose in addition to sucrose, glucose, and fructose in wucai was measured at 34 DAP and 46 DAP. We found that sucrose did not increase significantly, whereas glucose and D-galactose did more than fructose.”
Revised manuscript lines 449-507: revised to “AI promoted the hydrolysis of not only sucrose, but also raffinose and stachyose [26]. CWINV and VINV activities were positively regulated by their encoding genes and they all were the AI [26–28]. The downregulation of BFRUCT3 showed that sugar accumulation did not depend on the hydrolysis of sucrose in the vacuoles during wucai ripening. Thus, the up-regulated of CWINV4 during the ripening of wucai resulted in a significant increase in AI activity, allowing for more D-galactose and β-D-glucose production. Wucai leaf is both the source tissue and the sink tissue. We found that the expression of CWINV4 was significantly increased in wucai leaf compared to the other tissues at 46 DAP (Supplementary Figure S7). This result was contrary to that of Chinese cabbage [5,42]. The IM is the main tissue of sugar accumulation in Chinese cabbage. CWIN1 (CWINV), NIN-like (CINV), and VIN4b (VINV) had relatively lower expression in the inner leading leaves than the external leading leaves during Chinese cabbage ripening, especially in IM [5]. Three INV genes (encoding β-fructofuranosidase 1, β-fructofuranosidase 6, and β-fructofuranosidase 3) were also significantly downregulated in the inner leaves of yellow-head Chinese cabbage [42]. In addition, the basic leucine zipper (bZIP) transcription factor (TF) GmbZIP123 promoted the expression of three CWINV genes (CWINV1, CWINV3, and CWINV6) by directly binding to their promoters, resulting in higher levels of glucose, fructose, and sucrose in soybean [58]. A pitaya WRKY TF HpWRKY3 was associated with fruit sugar accumulation via the activation of the sucrose metabolic gene HpINV2 [59]. While there was no bZIP TF detected herein, WRKY TFs were detected in this study. Identifying which WRKY TFs can work with CWINV4 needs further analysis and verification.
The SPS activity did not change during the maturation of wucai, but SUS activity increased remarkably. Also, one DEG (SUS3) encoding SUS was up-regulated in the transcriptome data, and no SPS DEGs was found, consistent with the enzyme activities (Figure 8 B-C and Figure 5B). Therefore, it was inferred that SUS3 promoted the synthesis of sucrose to offset the hydrolysis of sucrose under CWINV4. Starch degradation during ripening is a key additional process for D-glucose accumulation in fruit and is catalyzed by the action of amylases [60]. The activity of AMY and DPE increased during mango ripening with a concomitant decrease in the starch content of the fruit [13]. BMY activity and BAMs (BAM1, BAM3, BAM3-like, and BAM5) were significantly down regulated (Figure 5B). DPE catalyzing starch conversion into D-glucose was also found to be downregulated (Figure 5B). However, there was no differential accumulation of starch during wucai ripening due to the downregulation of SS1 and SBE3 for starch synthesis. It follows that the accumulation of β-D-glucose did not originate from starch degradation during wucai ripening.
The cellulose hydrolytic enzyme beta-1, 4-endoglucanase (E1) gene, from the thermophilic bacterium Acidothermus cellulolyticus, was overexpressed in rice through Agrobacterium-mediated transformation [61]. Hydrolysis of transgenic rice straw yielded 43% more reducing sugars than wild-type rice straw [61]. It was found that overexpression of EG promoted the hydrolysis of cellulose, which is consistent with our study. Additionally, the up-regulated expression of BGL genes in a ripe rich-sugar mango variety showed that the genes could promote the accumulation of sugar [13]. There were no CBH DEGs detected in our transcriptome data (Figure 5B). However, we observed a significant increase in CL activity. It was inferred that CEL1, BraA03g023380.3C combined with BGLU16 catalyzed cellulose into β-D-glucose. A β-glucosidase from Clostridium cellulovorans (CcBG) was fused with cellulosomal endoglucanase CelD (CtCD) from Clostridium thermocellum [62]. CtCD CcBG showed favorable specific activities on phosphoric acid-swollen cellulose (PASC), with greater glucose production (2-fold) when compared with a mixture of the single enzymes, further supporting our conclusions [62]. The transcription levels in mature Chinese cabbage and rich-sugar mango were significantly higher than that of unmatured Chinese cabbage and low-sugar mango, which proved that the downregulated expression of HK led to the accumulation of more glucose [13,41]. Significantly reduced HK activity during maturation of wucai was accompanied by down-regulated expression of HK DEGs (BraA06g003260.3C, BraA08g002960.3C, BraA05g019040.3C, and BraA05g027230.3C), which reduced the loss of D-glucose and lead to more conversion of D-glucose to β-D -glucose. Similarly, downregulation of HK activity reduced phosphorylation of β-D-glucose, thereby promoting sugar accumulation.”
Revised manuscript lines 619-624: revised “Interferon related developmental regulator (IFRD) was mainly involved in plant salt tolerance, cold tolerance, and ABA signal transduction pathway in previous reports [63–65].”.
Revised manuscript lines 625-638: revised to “Some scholars have pointed out beneficial role of inositol in promoting sugar accumulation [66]. In the biosynthesis of inositol, the rate-limiting step is catalyzed by inositol-3-phosphate synthase (ISYNA) [67]. So, BraA01g000700.3C was speculated to be highly expressed after maturation to enhance sugar accumulation (Figure 7). S-adenosylhomocysteine hydrolase (SAHH) is a widespread enzyme in cells. Over-expression of SlSAHH2 could enhanced SAHH enzymatic activity in tomato development and ripening stages and resulted in a major phenotypic change of reduced ripening time from anthesis to breaker [68]. Interestingly, SAHH enzyme activity levels and SlSAHH2 transcript levels appeared to be inconsistent in some tissues. For example, SlSAHH2 were not significantly elevated in transgenic fruit, but its enzymatic activity remained at a high level [68]. From the above, it was assumed that SAHH2 decreased during the ripening process, but it still maintained a high level of enzyme activity to promote ripening and sugar accumulation in wucai.”.
Revised manuscript lines 413-427, 433-436, 508-616, 621-623, 640-657, 665-667: The content of the original manuscript was deleted.
Point 3. Descriptions/captions of figures could be made in larger fonts, and some diagrams should be enlarged (e.g. Figs. 2, 3, 5, or Supplementary. Fig.2).
Response: Thanks for your suggestion.
The pictures noted has been modified according to the suggestions.
Point 4. All abbreviations used should be explained, including those in figure captions.
Response: Thanks for your reminder.
All abbreviations used have been explained in the revised manuscript.
Q5. The Authors should check more carefully the Reference list, and improve it. In addition, there are small mistakes in the text of manuscript, including Reference list, that need to be corrected by Authors (e.g. lines: 22, 498, 703, 704, 768, 796, 846, 859, 862, 871, 877, 888, 898).
Response: Thanks for your reminder.
Due to the rearrangement of the introduction and discussion, the contents and numbers of the references have changed. Therefore, this document does not list the revisions of references. Please see the complete revised manuscript.
Revised manuscript lines 22: revised to “By orthogonal projection to latent structures-discriminant s-plot (OPLS-DA S-plot) and MetaboAnalyst analyses, D-galactose and β-D-glucose were identified as the major components of sugar accumulation in wucai.”.
Revised manuscript lines 528: revised to “300 μmol·m−2·s−1”.